# Does the Acknowledgement of αS1-Casein Genotype Affect the Estimation of Genetic Parameters and Prediction of Breeding Values for Milk Yield and Composition Quality-Related Traits in Murciano-Granadina?

**DOI:** 10.3390/ani9090679

**Published:** 2019-09-13

**Authors:** María Gabriela Pizarro Inostroza, Vincenzo Landi, Francisco Javier Navas González, Jose Manuel León Jurado, Amparo Martínez Martínez, Javier Fernández Álvarez, Juan Vicente Delgado Bermejo

**Affiliations:** 1Department of Genetics, Faculty of Veterinary Sciences, University of Córdoba, 14071 Córdoba, Spain; kalufour@yahoo.es (M.G.P.I.); juanviagr218@gmail.com (J.V.D.B.); 2Animal Breeding Consulting, S.L., Córdoba Science and Technology Park Rabanales 21, 14071 Córdoba, Spain; landivincenzo@yahoo.it (V.L.); amparomartinezuco@gmail.com (A.M.M.); 3Centro Agropecuario Provincial de Córdoba, Diputación Provincial de Córdoba, Córdoba, 14071 Córdoba, Spain; jomalejur@yahoo.es; 4National Association of Breeders of Murciano-Granadina Goat Breed, Fuente Vaqueros, 18340 Granada, Spain; j.fernandez@caprigran.com

**Keywords:** Restricted Maximum Likelihood, genetic parameters, genotype for αS1-CN, milk yield and composition

## Abstract

**Simple Summary:**

Genetic evaluations and the selection of breeding animals require the accurate estimation of genetic parameters for economically important traits. As a result, dairy livestock has evolved in response to the needs of producers and consumers. Genetic selection in goats has been mostly based on quantitative traits such as milk yield, fat, protein, and dry matter. However, as reported by the increased heritability values of these parameters after the inclusion of the different allelic variants of αS1 casein in evaluation models, the selection of animals carrying this gene could result in a more efficient genetic selection. High levels of genetic polymorphism (89.58% of polymorphic SNP—as only five out of the 48 SNPs assessed were monomorphic) that are related to greater production of coagulable proteins in milk, a fact that could be associated with a higher yield and improved curd firmness properties.

**Abstract:**

A total of 2090 lactation records for 710 Murciano-Granadina goats were collected during the years 2005–2016 and analyzed to investigate the influence of the αS1-CN genotype on milk yield and components (protein, fat, and dry matter). Goats were genetically evaluated, including and excluding the αS1-CN genotype, in order to assess its repercussion on the efficiency of breeding models. Despite no significant differences being found for milk yield, fat and dry matter heritabilities, protein production heritability considerably increased after aS1-CN genotype was included in the breeding model (+0.23). Standard errors suggest that the consideration of genotype may improve the model’s efficiency, translating into more accurate genetic parameters and breeding values (PBV). Genetic correlations ranged from −0.15 to −0.01 between protein/dry matter and milk yield/protein and fat content, while phenotypic correlations were −0.02 for milk/protein and −0.01 for milk/fat or protein content. For males, the broadest range for reliability (R_AP_) (0.45–0.71) was similar to that of females (0.37–0.86) when the genotype was included. PBV ranges broadened while the maximum remained similar (0.61–0.77) for males and females (0.62–0.81) when the genotype was excluded, respectively. Including the αS1-CN genotype can increase production efficiency, milk profitability, milk yield, fat, protein and dry matter contents in Murciano-Granadina dairy breeding programs.

## 1. Introduction

The Murciano-Granadina goat is the most popular breed in Spain and it is distributed across several countries in Europe, Africa, and South America [1]. The breed is generally maintained in extensive, semi-extensive, and intensive systems, and in very different climatic conditions [2], accounting for an average yield per standardized lactation of 210 days of 416.0 ± 197.9 kg for milk, 19.54 ± 7.12 kg for fat, 13.25 ± 5.31 kg for protein, and 52.33 ± 17.16 kg for dry matter [3]. However, the productivity of the Murciano-Granadina goat has been reported to reach much higher levels, as indicated by yield records of more than 1500 kg [4]. 

The production of goat derived products, such as milk, cheese, and meat, has been increased by 66% worldwide over the last 20 years [5] as a result of consumers’ preferences. Not only have goats gained prominence in smallholder farming systems [6], but consumers are also becoming more and more attracted to the nutritious health-beneficial properties of goat’s milk such as its application as replacement milk in food allergy cases [7].

Genetic selection has contributed to the development and advancement of the dairy sector, focusing on the improvement of economically important aspects associated with increased productivity and profitability [8]. The specific characteristics considered for the selection of dairy cattle populations have evolved in response to changes in the needs of producers, consumers and the society, with the help of advances in technology and data recording programs, among others [9].

The estimation of the genetic parameters, and knowledge of the components of (co)variance and estimates of heritability for milk production and quality traits, result in the selection of animals with superior genetic merit to enhance the selection and breeding of dairy traits as desired [10]. Such selection implies the estimation of the genetic and phenotypic parameters, the choice of selection criteria for each production system, and the economic ponderation of each of the selection criteria influencing profit in these production systems [11,12].

The estimation of genetic parameters for the milk yield and milk components of goat breeds has been carried out in different breeds in many countries across the world [13]. Estimates of genetic parameters for dairy goats are available from Latin American and Mediterranean countries [10,14], but also from other populations in New Zealand, Norway, and South Africa [15,16,17]. 

Genetic factors comprise elements involving high variability among individuals. Among them, there is a constant concern in researchers about how αS1-Casein genetic variants may affect milk components and their profitability, even when the first findings date back to decades ago [18]. However, the study of caseins in goats remains complex due to the extensive polymorphism in the four existing casein loci [19]. 

The *CSN1S1* gene has a transcriptional unit of 16.5 kb in length, comprising 19 exons, which vary in length from 24 pb to 358 pb [20] and 18 introns [21]. So far, more than 16 alleles have been detected and grouped into four classes, according to the different levels of expression of αS1-CN in milk. ‘Strong’ variants (A, B1, B2, B3, B4, C, H, L, and M) produce approximately 3.6 g of αS1-CN per liter of milk; The ‘medium’ variants (E-I) produce 1.6 g of α S1-CN, the ‘weak alleles’ (D, F, and G) produce 0.6 g of α S1-CN in small ruminants’ milk, particularly of goats [22] and the ‘null’ alleles (01, 02 and N) result in the absence of αS1-CN in milk [23]. Phenotypically, α S1-CN has been reported to account for 26.43% of total protein percent at 210 days lactations in Murciano Granadina goats [24].

Several studies have analyzed the effects of the polymorphism of casein genes on dairy yield and milk quality in different goat breeds [25]. These authors have revealed that polymorphisms in the *CSN1S1* locus have significant effects on casein content, total protein content, fat content and the technological properties of milk.

Hence, the main objective of this study is to comparatively estimate genetic parameters and breeding values of milk yield and components (protein, fat and dry matter in kg), using two models, one including the αS1-CN genotype as a biomarker fixed effect, and the other excluding it. This procedure was carried out to analyze the potentially occurring discrepancies or similarities obtained after both models, assessing whether the inclusion of αS1-CN genotype may result in improved efficiency and profitability of the selection techniques applied in the breeding programs for milk production and quality in Murciano-Granadina goats.

## 2. Materials and Methods 

### 2.1. First Phase: Study Sample Selection Process

During this initial phase, the statistical properties of the variables of milk yield, fat, protein, and dry matter contents (kg) from the Murciano-Granadina goat breed’s complete historical routine milk record until 2017 were evaluated (*n* = 2,359,479 registries belonging to 151,997 animals born from 1998 to 2017). This whole routine milk recording set was submitted to a depuration process to rule out those goats for which registries fell outside the Murciano-Granadina goat breed’s reference values for milk yield and composition (protein, fat and dry extract). After this depuration process, the pedigree matrix considered in our study for genetic analyses consisted of 29,397 animals (26,993 does and 2404 bucks) born from 1998 to 2017, registered in the historical pedigree, accounting with direct records in the relationship matrix and at least connected via one known ancestor. Then, parametric assumptions were tested (normality, homoskedasticity, sphericity, multicollinearity, and outlier presence) within the raw phenotypes. 

After this, parametric assumptions were retested on females and clustered by partition number for every fixed effect in our model (farm, birth year, month and season and birth type). This was performed as a way to adjust phenotypes and determine whether the lack of normality could be attributed to the properties of the sample studied as a normal distribution could be expected for productive data from goats at the same lactation status. 

To achieve this objective, SPSS Statistics for Windows’, Version 24.0, IBM Corp., Armonk, New York (2016) split file routine was used. Both procedures (complete historical and adjusted data, respectively) reported the violation of the parametric assumptions already reported, a fact that suggested the application of a nonparametric approach. After parametric assumptions had been tested, we selected our sample set comprising 710 goats (with 2090 records collected during the years 2005–2016) and genotyped their αS1-CN background to evaluate the influence of such a genotype following the sample selection criteria described in Pizarro et al. [26] Animals were ranked considering their predicted breeding values (PBVs) for milk yield, fat and protein and the relationships among these traits at a previous genetic evaluation, using the combined selection index (ICO) procedures defined in Pizarro et al. [26]. The 236 females presenting the lowest ICO values in the rank, 238 females with values around percentile 50, and the 236 females presenting the highest ICO values in the rank were chosen, so as to perform an adjusted representative sampling of the genotype distribution in the population). Univariate (to estimate heritabilities) and bivariate analyses (to estimate genetic and phenotypic correlations) were performed separately for the selection of the animals that could be genotyped, aiming at preventing the incorrect estimation of heritabilities as an effect of the epistasis between traits. Then, we applied mixed model procedures using an animal model (BLUP) by restricted maximum likelihood, with the MTDFREML software package [27]. We followed an interaction process seeking the achievement of convergence criteria levels of 10^−12^. The genetic evaluation involved those goats that presented direct records and were at least connected via one known ancestor within the relationship matrix. This matrix consisted of the historical pedigree until 2017 (151,997 females). After we achieved convergence, predicted breeding values were computed with the MTDFREML software.

To rank the goats in the relationship matrix and select the ones to be genotyped out of them, we computed a combination of goat’s predicted breeding values for three of the four traits measured (i.e., milk yield, fat and protein contents) using a combined selection index (ICO) procedure as described in Van Vleck [28]. Dry matter was not included in the computation of the combined index (ICO) given the potential redundancy occurring as it comprises an indirect measurement of fat and protein components. The procedure to set the combined selection index used and the premises considered to determine it are described in Pizarro et al. [26].

The model and methods used at the preliminary genetic evaluation were designed after a routine process including commonly reported effects for the evaluation of milk yield, protein, fat and dry matter components, via the common parametric approach addressed by several authors [29,30]. As parametric approaches had been used to test these preliminary model and methods, we may be able to infer a comparison of the results obtained after running nonparametric tests on our data consisting of variables that are commonly studied using parametric approaches. 

#### 2.1.1. Study Sample

After the depuration process and sample selection for genotyping had been accomplished, we retained registries collected during the period from 2005 to 2016 from 710 Murciano-Granadina non neutered lactating goats present in the studbook. The age of goat in the sample ranged from a minimum of 14.03 to a maximum of 130.13 months old. As age did not distribute normally (*p* < 0.001 for Shapiro-Wilk Francia’s tests to test for normality in samples up to 5000 cases), we provide Q1 (17.70 months), median age (28.77 months) and Q3 (41.10 months). The properties of parturition age across number of parturitions were assessed through Shapiro-Wilk Francia’s W test, Kurtosis and Skewness revealing the data distributed normally (*p* > 0.05 for Shapiro Wilk Francia’s W test, around 3 for Kurtosis and a value ranging from −0.5 to 0.5 for Skewness).

#### 2.1.2. Lactation Standardization

Regular management policies normally followed and applied at Murciano-Granadina farms rely on two kidding seasons per year (polyestric breed), which translates into economically important income derived from no longer than 210–240-day lactation periods [31]. For the analyses carried out in the present study, total milk yield and content were estimated until 210 days of lactation as suggested by Brito et al. [29] and expressed in kg following the procedure described in Pizarro et al. [26].

#### 2.1.3. Milk Composition Evaluation

The amount of milk collected during one milking period was determined for the 710 goats individually after typifying lactation period at 210 days, following seven controls after parturition carried out periodically every 30 days. Samples were then sent to the official Milk Quality Laboratory (Córdoba, Spain) for composition determination (total fat, protein and dry matter) using a MilkoScan™ FT1 analyser. Data was then purged and 2090 registries form 710 goats were considered at the statistical analysis. Appendix A
Appendix A describes the individuals and observations and the genotypic frequencies of each CSN1S1 genotype in Murciano-Granadina goat for the sample used in our study and for population frequencies presented at a previous paper [32].

### 2.2. Second Phase: Determining the Effect of Genotype

During this phase, we adjusted phenotypes and checked potential intergenotypic differences, following nonparametric statistical procedures, as described in Pizarro et al. [26].

#### DNA Bank, αS1-Casein (CNS1) Genotyping and Mutation Identification

The procedure of DNA sampling, αS1-casein (CNS1) genotyping and mutation identification was carried out following the premises described in a previous study by Pizarro et al. [26]. DNA information from 117,225 blood samples belonging to Murciano-Granadina breeding males and females stored in the Germplasm Bank held by Animal Breeding Consulting, Inc., Córdoba (Spain) Samples were owned by the National Association of Breeders of Goats of Murciano-Granadina Breed (CAPRIGRAN). When the blood sample for a certain animal registered in the studbook was not stored at the germplasm bank, such animal was not considered to comprise the study sample.

The encoding sequences of the allelic types of the αS1-Casein gene reported by Genebank are AJ504710, AJ504711, AJ504712, and X56462, respectively, corresponding to the alleles identified in our study (F, N, A, and B, respectively) (Appendix A
Appendix A). Preliminary statistical analyses reported the presence of a 457 bp LINE element determining the presence E allele, involving and excluding 11 bases as described in Appendix A
Appendix A.

PCR procedures were used to search for the alleles reported in Maga et al. [33] in order to identify the mutations in the samples in our study, as reported by Pizarro et al. [26].

### 2.3. Statistical Analysis of Non-Genetic and Genetic Factors

We analysed the effects of categorical factors (farm, parturition year, month and season, kidding type and genotype) on continuous variables (milk yield, protein, fat and dry matter (expressed in kg). We did not transform data given the analyses were performed on normalized milk yield expressed in kg. After this, the total number of kg of normalized milk yield at 210 days was used to compute the amount of each component expressed in kg by multiplying this total number of kg by the percentage of fat, protein and dry extract determined at the laboratory. Although the additive component could not be considered to be negligible for milk yield and composition, mean inbreeding in the historical pedigree of the breed until 2017 (*n* = 2,359,479 registries) is as low as 0.29%, thus irrelevant and not included in the model as described in Pizarro et al. [26]. The statistical procedures described by these authors [26] were followed as a way to counteract the obstacles represented by the effects of the properties of the data from milking records as already described in a previous paper [34]. Parametric assumptions were tested on the whole milking record available for Murciano-Granadina goat breed (*n* = 2,359,479) to determine the statistical approach to follow seeking the best validity for results. As normality assumption was violated (Kolmogorov Smirnov, *p* < 0.001, highly positively skewed and platykurtic distribution) for milk yield, fat, protein and dry matter content expressed in kg (Appendix A) a non-parametric approach was followed. Homoscedasticity (Levene’s test, *p* < 0.001) and Mauchly’s W test of sphericity (Mauchly’s W = 0.29), χ^2^(5) = 8,327,656.071, *p* < 0.001) were neither fulfilled. These assumptions were retested to examine whereas data could presumably follow a normal distribution when the information of goats at the same lactation status is considered. Additionally, the statistical parametric assumptions were also tested on our sample. Normality was only slightly detected (Kolmogorov-Smirnov, *p* < 0.05), for goats at the fourth to the sixth lactation, which in turn did not influence milk yield, protein, fat and dry matter. Contrastingly, no evidence of normality was found for any of the lactation stages from first to sixth lactation, with evidences of heteroscedasticity and lack of sphericity, hence variance calculations may be distorted, which would result in an inflated F-ratio. Normality tests were carried out with sfrancia routine of StataCorp Stata version 14.2 evidencing a highly statistically significant (*p* < 0.001) deviation of normality in the traits studied. Given that genotyped goat number was a limiting factor, we also tested for homoscedasticity between genotype variants in this smaller set with Levene’s test, evidencing the occurrence of heteroscedasticity as well (*p* < 0.05). Despite univariate outliers could be the base for the violation of the assumption of normality for the data related to milk yield and composition traits in our population, we decided not to discard them as even when these values are over the values that are commonly reported for the breed in literature, they are found within the limits for empirical data field observations. For instance, the fact that the Murciano-Granadina breed presents a high frequency of goats producing up to 1300 L of milk in 210 days of lactation in routine performance checks. Highly statistically significant (*p* < 0.01) multicollinearity assumption among variables was evidenced after the results of Spearman rho (ρ) correlations. All these pieces of evidence determined our choice to perform nonparametric tests to statistically assess the milk yield and composition data recorded. Hence, a Kruskal-Wallis H was performed to test for interlevel differences. A summary of the levels per factor considered in the model of study can be consulted in Table 1.

After Kruskal-Wallis H, the power of the factors on milk yield and composition traits was computed via the determination of partial eta squared (ηp^2^) coefficients [35] (Table 1). Then, Dunn’s test assessed the differences in the distribution of each of the traits considered across the groups of the same factor affecting them. The Bonferroni correction was applied to compensate for Type error I increase derived from the high number of factors considered. An independent-sample median test was performed to corroborate previous results analysing the median across levels of the same factor. The sfrancia routine of StataCorp Stata version 14.2. was used to perform Shapiro Wilk Francia’s tests. All nonparametric tests were carried out using the independent samples package from the non-parametrical task of SPSS Statistics for Windows, Version 24.0, IBM Corp. (2016). Further details for this statistical analysis can be found in Pizarro et al. [26].

### 2.4. Genetic Analyses: Genetic Model Comparison, Phenotypic and Genetic Parameters and Predicted Breeding Value Estimation

#### 2.4.1. Genetic Model Comparison, Phenotypic and Genetic Parameter Estimation

This work was carried out with data made available by the Murciano-Granadina Breeders Association, collected in the framework of its breeding program. Overall, the pedigree matrix included records of 29,397 animals with direct records related through at least one known ancestor. (26,993 does and 2404 bucks), while the phenotype data set included 2090 records of 710 goats evaluated between 2005 and 2016. Therefore, a multitrait animal mixed model with repeated measures was used to estimate (co) variance components, and the corresponding heritability, repeatability, phenotypic and genetic correlations and standard errors of such correlations for the traits under examination. In matrix notation, the following multitrait animal model with repeated measures was used:

Yijklmnopq = μ + Fari + Yeaj + Monk + Seal + Typm + Genn + b_1_Ao + $b_{2}^{2}A$o + Animalp + PEq + eijklmnopq, where Yijklmnopq is the separate score of milk yield and components (fat, protein and dry matter in kg) for a given animal; μ is the overall mean; Fari is the fixed effect of the ith farm/owner (i = 59 farms); Yeaj is the fixed effect of the jth year of parturition (j = 2005–2016); Monk is the fixed effect of the kth month of parturition (k = May to December); Seal is the fixed effect of the lth season of evaluation (j = Autumn, Winter, Summer, and Spring); Typm is the fixed effect of the mth type of birth(m= 1 to 5 kids); Genn is the fixed effect of the nth Genotype (*n* = AA, BA, BB, AE, BE, and EF); age in months was considered a linear and quadratic covariate, hence b_1_ and $b_{2}^{2}$ are the linear and quadratic regression coefficients on the age of evaluation (Ao), Animalp is the random additive genetic effect of the pth goat, PEq is its permanent environmental effect, and eijklmnopq is the random residual effect. At a second stage, genotype was not considered in the model to isolate the possible effects derived from considering its effects on the traits measured. In matrix notation, the multitrait animal model with repeated measures excluding αS1-casein genotype used was: 

Yijklmnop = μ + Fari + Yeaj + Monk + Seal + Typm + b_1_An + $b_{2}^{2}A$n + Animalo + Pep + eijklmnop, where Yijklmnop is the separate score of milk yield and components (fat, protein and dry matter in kg) traits for a given animal; μ is the overall mean; Fari is the fixed effect of the ith farm/owner (i = 59); Yeaj is the fixed effect of the jth year of parturition (j = 2005–2016); Monk is the fixed effect of the kth month of parturition (k = May to December); Seal is the fixed effect of the lth season of evaluation (j = Autumn, Winter, Summer, and Spring); Typm is the fixed effect of the mth type of birth (m= 1 to 5 kids); age in months was considered a linear and quadratic covariate, hence b_1_ and $b_{2}^{2}$ are the linear and quadratic regression coefficients on the age of evaluation (An), Animalo is the random additive genetic effect of the oth goat, PEp is its permanent environmental effect, and eijklmnop is the random residual effect. 

MTDFREML software package [27] was used to perform Restricted maximum likelihood approach-based univariate analyses in order to compute heritabilities and variance components. The same software was used to carry out bivariate analyses to estimate covariates and genetic and phenotypic correlation. The iteration process used sought a convergence criterion level of 10^−12^. Link functions can be found in Boldman et al. [28]

#### 2.4.2. Non-Genetic Factors Estimation (BLUES) and Breeding Value Prediction (BLUPS, PBVs)

After convergence was reached, we directly estimated non-genetic factors estimators through best linear unbiased estimators for fixed effects (BLUES) and predicted breeding values through best linear unbiased predictors for random effects (BLUPS, PBVs), their accuracies and reliabilities for milk yield, fat, protein, and dry matter for each animal in the matrix, using the MTDFREML software [27].

Standard error of prediction (SEP), reliability (R_AP_) and accuracy (RTi) do not provide basically the same information. They differ from the very definition and equation determination (R_AP_ = RTi^2^ = (1–SEP^2^/Va)^2^), from which Va is genetic additive variance. On the one hand, reliability could be described as the likeliness of someone repeating the experiment and getting the same result (repeatability), whereas accuracy is how close your value is to the real value, hence values should be interpreted differently. According to the international beef recording scheme [36], the following guide may assist as a rule of thumb to interpret accuracy (RTi). Less than 50% accuracies mean PBVs are preliminary, thus calculated basing on little information and hence very prone to change substantially as more direct information on the animal becomes available. Accuracies ranging from 50–74% accuracy (medium) suggest that PBVs may have been calculated based on the animal direct information and some limited indirect pedigree information. Medium/high accuracies are denoted by 75–90% and may be calculated considering the animal’s direct information together with the performance of a small number of its offspring. Accuracy values over 90%, report estimates of the animal’s true breeding value, as it unlikely that PBVs will change considerably even if additional information from offspring is added. Regarding reliability, the rule of thumb proposed by KWPN (Royal Dutch Sport Horse) [37] is as follows; values less than 30% are generally unreliable, 30–55% poor reliability, 55–65% sufficient reliability, 65–75% more than sufficient reliability, 75–90% good reliability, >90% very reliable and repeatable with values around 60% meaning the information strongly relies of offspring information, what would not be desirable. On the other hand, the standard error of prediction (SEP) measures how large prediction errors (residuals) are for your data set measured in the same unit as your variable, hence provides a direct measure of possible change, that is the risk of the true breeding value of animal (TBV) not to be centered on the PBV. According to Van Vleck [38], possible change is risk in units of the trait and can be ‘positive’ or ‘negative’, what means the chance true BV may exceed PBV by a certain amount (possible gain) is the same as the chance true BV is less than PBV by the same amount (possible loss). In this context, confidence ranges are frequently used to determine probabilities of possible change assuming a normal distribution of TBV around the PBV. One-half of TBV would be expected to be greater than the PBV and one-half would be expected to be less than the PBV. The interval from PBV–(1)SEP to PBV + (1)SEP corresponds to 68% of likelihood that the TBV for an animal is centered on the PBV for the animal. The range can be narrowed or widened corresponding to the probability of TBV in the interval. For example, the interval from PBV–(2)SEP to PBV + (2)SEP would be expected to contain 95% of TBV. Units of SEP other than (1) or (2) would correspond to other confidence ranges. With a 68% confidence range, 32% would be half over and half below the ranges’ ends, while with the 95% range, the percentage falling out of the range would be 5% (again half over and half below each end, respectively). Ranges for many combinations of PBV and SEP may overlap considerably. Then, by observing which PBV centers the range and comparing ranges, we may obtain a more direct measure of risk than that from accuracy (RTi).

#### 2.4.3. Model Comparison

Descriptive statistics were evaluated to study the distribution properties of predicted breeding values obtained with the two models (including and excluding genotype for αS1-CN as a fixed effect). Then, we computed and compared the linear regression equations described by the predicted breeding values for milk yield and components obtained after both models to determine the possibility of excluding genotype from the routine genetic evaluation of the breed given the costs involved in genotyping). Pearson product-moment correlation analysis between the predicted breeding values (PBV) for milk yield, protein, fat and dry matter (expressed in kg) obtained using both the model including the αS1- casein genotype and the one excluding it for all the animals included in the pedigree (*n* = 29,397) of Murciano-Granadina breed goats was carried out to check for the replicability of the results.

### 2.5. Ethics Committee Statement

The study followed the premises described in the Declaration of Helsinki. The Spanish Ministry of Economy and Competitivity through the Royal Decree-Law 53/2013 and its credited entity the Ethics Committee of Animal Experimentation from the University of Córdoba permitted the application of the protocols present in this study as cited in the fifth section of its second article, as the animals assessed were used for credited zootechnical use. This national Decree follows the European Union Directive 2010/63/UE, from the 22nd of September of 2010. Furthermore, the present study works with records rather than live animals directly, hence no special permission was compulsory.

## 3. Results

### 3.1. Statistical Analysis of Non-Genetic and Genetic Factors

A summary of the descriptive statistics for the fixed effects and covariates comprising the model testing for Murciano-Granadina goat milk yield and components is shown in Appendix A
Appendix A. All traits significantly violated normality assumption as suggested by the results of Kolmogorov-Smirnov Test (*p* < 0.001), kurtosis deviation (0.871 to 1.191) and skewness values (1.218 to 2.897) (Appendix A
Appendix A). Percentage of components (μ ± SD) in our milk samples were 5.213% ± 1.021%; 3.544% ± 0.557%; 14.308% ± 2.042%, for fat, protein and dry matter respectively. A summary of the output for the Kruskal Wallis H test and partial eta squared coefficient (ηp^2^) is reported in Table 1.

MANOVA (as the design followed in our model) involves the use of non-independent or repeated measures while ηp^2^ quantifies for intergroup differences in our MANOVA plus their associated error variance expressed as a percentage. Then, univariate follow-up tests must be carried to further isolate the groups for which such significant and mean differences can be found. The literature recommends the use of partial eta square instead of eta square after a multifactorial design has been approached, given partial eta square reports an index of association strength between independent variables and dependent factor excluding the variance produced by the rest of factors considered in the model [39]. This relies on the fact that, as the Kruskal-Wallis test statistic is based on a single independent factor, no other factor accounts for the variance explained by this factor itself in the dependent variable, ηp^2^ equals η^2^, which is of remarkable importance when we consider non-possibly overlapping empirical variables, such as the ones in this study. 

A complete description of the statistical analyses carried out on the nongenetic and genetic fixed effects included in the genetic model can be found in Pizarro et al. [26].

### 3.2. Genetic Analyses

#### 3.2.1. Genetic Model Comparison, Phenotypic and Genetic Parameters Estimation

Estimates of non-genetic fixed effects (BLUES) obtained from the REML quantitative genetic analysis, including age as a linear and quadratic covariate, the fixed effects of farm/owner, birth year, month and season, and birth type, including and excluding genotype from the model are shown in Appendix A
Appendix A. The estimates for heritability, genetic, phenotypic and environmental variance obtained through REML methods for both model including and excluding the genotype as a fixed effect are shown in Table 2. The genetic (r_G_) and phenotypic correlations (r_P_) estimated are shown in Table 3. 

#### 3.2.2. Breeding Value Prediction and Comparative Descriptive Analysis

A summary of the descriptive statistics of predicted breeding values (PBV), standard error of prediction (SEP), accuracy (RTi) and reliability (R_AP_) for the milk yield and components sorted by sex is shown in Table 4. Maximum and minimum PBVs for all traits were slightly to moderately higher in bucks when genotype was excluded than when it was excluded except for fat. By contrast, maximum and minimum PBVs were slightly higher for milk yield and fat, while they were slightly lower for protein and dry matter when genotype was excluded. The broadest range for R_AP_ was reported for bucks (0.45 to 0.71) when compared to does (0.37 to 0.86) when the genotype was included, while the values for the same range remained similar (0.61 to 0.77) for bucks and does (0.62 to 0.81) when genotype was excluded. The lowest R_AP_ was reported for the PBVs for dry matter when genotype was included while the highest was reported for fat under the same premises. Contrastingly, RTi scores, range from 0.67 to 0.84 when genotype is included and from 0.78 to 0.88 when genotype is excluded for bucks, and from 0.61 to 0.93 when genotype is included and from 0.79 to 0.90 when genotype is excluded for does, respectively. 

Table 5 shows the results for Pearson product-moment correlation analysis between the predicted breeding values (PBV) for milk yield (kg), fat (kg), protein (kg) and dry matter (kg) obtained using both the model including the αS1- casein genotype and the one excluding it for all the animals included in the pedigree (*n* = 29,397) of Murciano Granadina goat breed. Regression equations for the predicted breeding values for milk yield, and components obtained after both models to determine the possibility of excluding genotype from the routine genetic evaluation of the breed given the costs involved in genotyping, are shown in Figure 1.

## 4. Discussion

The evaluation of potentially occurring discrepancies or similarities between the two models included in our study may outline the benefits derived from the inclusion of αS1-CN genotype, considering that genotypic variants could result in an improved efficiency of the selection techniques applied. Then, conjoining the comparison of genetic parameters and predicted breeding values, with the study of the economic repercussion of the relationship and interaction between milk component traits (linear and nonlinear) [40], especially those which may act as relevant predictors of quality in cheese or yoghurt production, the profitability of dairy goat breeding programs can be optimized [41]. 

Schmitz et al. [41] explained that the advantage of multivariate analyses is based on taking into account not only the expectations of covariates within the pair, but also the information that relates the traits to each other and that of the different observations for the same individual. Through this additional information, the power increases to detect unrelated genetic deviations in cases where there is shared environmental variation. Shared environmental correlations not only help in the detection of joint environmental variance but also heritability. In the same way, phenotypic correlations also contribute to the detection of genetic or shared environmental variance in the multivariate case, albeit on a smaller scale. The only situation in which there will be no increase in power with respect to a univariate case is when the phenotypic correlation between the variables is non-existent. Thus, Neale et al. [42] pointed out that the power of twin studies to detect variations of the genetic additive and environmental components depends on factors such as the effect considered and its potency on the studied variable.

New theoretical and computational developments allow the use of models that better describe biological processes, and as a result, they report predictions of genetic values and estimations of increasingly reliable variance components [43], as they progressively minimize the estimation errors associated with the sample structure and the estimation methodology used [44]. 

There is a large amount of variation in the literature with respect to estimates of heritability of milk yield and their components (fat, protein and dry matter) in goats [8]. Heritability values for the production of milk, fat, protein and dry matter content expressed in kg when the genotype factor of αS1-CN was considered as fixed effect were 0.40, 0.29, 0.53 and 0.16, respectively. On the other hand, when this factor was excluded, they were 0.40, 0.31, 0.30 and 0.15, respectively (Table 1).

Analla et al. [45] reported heritabilities which were around half of those reported in our study (0.18, 0.16, and 0.25 with single-trait analysis) for milk yield, fat content, and protein content for the same breed, respectively. However, despite genetic correlations between fat and protein content were similar (above 0.90), phenotypic correlations were moderately lower than those reported in our study. Contrastingly, Genetic correlations between milk yield and fat or protein were negative and moderately high. The same study by Analla et al. [45] reported phenotypic correlations between milk yield and protein and fat content that were negative but moderate as well. These negative correlations have been widely reported to be a result of the procedure followed to quantify milk composition using percentages or fractions (g/kg). Our higher heritabilities and lower correlations match some of those found in the literature and were maintained within the ranges of values estimated in other goat populations for milk yield (0.17 to 0.41) [46,47,48]; fat (0.26 to 0.62) [48], protein (0.14 to 0.67) [43,49], and dry matter (0.16 to 0.36) [50]. Estimates of heritability confirm the existence of a considerable level of genetic diversity that allows the possibility to continue improving the production of milk, fat, protein and dry matter in breeding programs and selection of Murciano Grandina breed goats, even if they are highly selected breeds [26,47,49,51], such as the one in our study.

The inclusion of the genotype of αS1-CN in the model reported slightly higher heritability values for protein and dry matter content while the values of the phenotypic correlation between the same variables were slightly lower when the genotype was considered in the model. Estimates of genetic parameters, and among them of heritabilities, of traits can vary considerably between studies because of the breed involved, the population sampled, environmental conditions of the study, and random and systematic errors in the estimation process or measurement [52]. 

Heritability standard errors for the production of milk, and fat, protein and dry matter content including the αS1 casein genotype as a fixed effect (0.06; 0.05; 0.02 and 0.04, respectively) and for the production of milk, and fat, protein and dry matter content without the inclusion of the genotype αS1 casein as a fixed effect (0.07; 0.05; 0.03 and 0.05, respectively) (Table 1) were kept within the ranges of estimated values in other populations of dairy goats (0.001 to 0.07) [51], indicating that the estimation of the parameters studied in our samples are reasonable.

The estimation of the coefficient of genetic and phenotypic correlation is of great importance in the implementation of selection processes since it provides an overview of the possible proportion of genes that may cause variation in a population in two or more independent characters simultaneously [53]. In our study, genetic correlations (Table 2) including and excluding the genotype of αS1-CN were negative between milk yield and protein content (−0.01), fat and dry matter content (−0.09), milk yield and fat or protein content (−0.01) and protein and dry matter content (−0.15). 

Phenotypic correlations including and excluding αS1 casein genotype were −0.02 for milk yield and protein content and −0.01 for milk yield/fat and protein content, respectively. Very similar values were found in Saanen and Alpina breeds [48]. These negative values have been reported for genomic correlations between loci and have been attributed to the negative linkage disequilibrium expected for traits under directional selection [54], as would happen in our study. The negative genetic and phenotypic correlations between milk yield and fat, protein, and dry matter content revealed unfavorable associations, which could adversely affect the quality of various dairy products especially those derived from the cheese industry. Therefore, dairy goats must be selected using indices not only considering the relations among these traits but also including αS1 casein genotype provided its association with cheese yield, and a firmer curd [55], as this inclusion may maximize the economic response from the markets commercializing these goat milk-derived products.

The values for genetic correlations showed a positive and slightly variable value from low and moderate both when the genotype of the αS1-CN was considered in and excluded from the model. It is worth highlighting, the genetic correlation between milk yield and fat content and milk yield and dry matter content of 0.01 when the genotype was included. This correlation was -0.01 and 0.00 when on the contrary the genotype was excluded. Also, the highest genetic correlation was the one between the protein and fat content, 0.93 and 0.97, when the genotype was excluded and included, respectively, followed in descending order by the correlation between protein and dry matter (0.14 and 0.23, when the genotype was included and excluded, respectively). The genetic correlations between milk yield and protein content presented the values of −0.02 and −0.01 when the genotype was included and excluded, respectively.

For their part, phenotypic correlations, including and excluding the genotype of αS1-CN reported similar values between the fat and protein components (0.97 and 0.93, when the genotype was excluded and included, respectively) which were within the ranges estimated by some authors in other dairy goat populations [17,56,57,58,59,60,61].

However, it is worth noting that, although there is an almost total similarity between the values of genetic correlations in both models, which would be expected since we evaluated the same traits in the same population set, phenotypic correlations showed to be moderately different for the rest of possible combinations of traits when the genotype for αS1-CN was included and excluded. The basis for these differences could be based on the explanatory significant linear effect exposed for the genotype, from 8.3% to 9.2%, for the contents in fat and protein, respectively in the categorical regression models previously analyzed. The positive correlations found for fat and protein production could produce an increase in the economic value for this dairy breed, as has been observed for other breeds whose milk is used primarily for cheese production, since cheese production depends on the total amount of protein and fat produced per year per goat to a large extent [46].

Verdier-Metz et al. [62] studied cheese performance calculating it as fresh performance by dividing the weight of fresh curd by the quantity of milk used for cheesemaking and dry matter performance by multiplying fresh performance by molded curd’s dry matter value. A wide range of values (55 to 85 g/kg) was reported by these authors for fat and protein ratio between manufactured kinds of milk. The same ratio linearly accounted for 77% of fresh yield and 87% of dry matter yield variability. This suggests higher fat/protein ratio values may contribute to an improved cheesemaking and curd formation performance. Pizarro et al. [26] suggested that such higher fat/protein ration could be possibly reported in goats presenting AE hybrid allelic combinations, as suggested by their findings regarding fat and protein contents being over the median (21.580, 14.878, and 59.808 kg for fat, protein, and dry matter content, respectively).

The accuracy and reliability parameters for the breeding values for milk yield including the genotype of αS1 casein were slightly lower than those reported by the model excluding the genotype for αS1 casein, for milk yield in males in the range of 0.06–0.10 (RTi, R_AP_ respectively). However, for females in both models, excluding and including the genotype effect, the range was 0.04-0.07 higher for the model excluding the genotype of αS1 casein.

On the other hand, the inverse situation occurs for fat content, for which the reliability for breeding values increased in the range of 0.02–0.04 (RTi, R_AP_ respectively) for males and 0.08–0.14 for females (RTi, R_AP_ respectively) when the genotype is included as a fixed effect. The values for R_AP_ range from poorly reliable to very reliable and repeatable. This is supported by RTi scores, which suggest that PBVs may have been calculated based on the animal direct information and some limited indirect pedigree ancestral information or belonging to a small number of its offspring. For instance, the differences found could be attributed to the fact that although direct observations from females were available, from males they were not, as these are unable to produce milk, what compelled us to evaluate them indirectly through the records from the females that were genealogically related to them. This has been suggested to occur in small samples, as the accuracy of a male’s breeding values increase as more of his daughters are measured and when further male genomic reliable information is not available [63]. The same situation is described for the predicted breeding values of protein content as their accuracies and reliability were 0.06–0.10 (RTi, R_AP_ respectively), for the males and 0.13–0.23 (RTi, R_AP_ respectively) for females, when the genotype of the αS1 casein was included as fixed effects in the model. On the contrary, the dry matter describes the same trend as in the production of milk, with accuracies and reliability increasing in the range of 0.11–0.14 (RTi, R_AP_ respectively) for males, and 0.18–0.21 (RTi, R_AP_ respectively) for females when the genotype of αS1 casein was not considered as a fixed effect in the model. 

Some authors have reported the possibility of accuracies to reach values close to zero or even zero for distant or unrelated animals when genetic analyses are performed on deep pedigrees (that is considering a large number of generations, involving a great number of unknown genetic background animals -founders-) as the one in our study (from 1 to 15 depending on the animal). Clark et al. [64] calculated accuracy as the correlation between estimated and true breeding values for 250 animals with no phenotype ranging from one to eight generations (depending on the individual). These authors did not find significant differences in accuracy between shallow or deep pedigree BLUPS when the animals in the pedigree had a close relationship. However, when a shallow pedigree was used, and animals were distantly related (maximum relationship ranging from 0 to 0.125) or unrelated (no pedigree relationship), all breeding values estimated were zero and accuracies close to zero (average relatedness in our pedigree ranged from 0% to 1.07%). In contrast, when pedigree was deep, breeding values were predicted with a significant accuracy when there was a distant relationship between animals and accuracy reduced to very close to zero when animals were unrelated, what may support our findings (Table 4). Genetic evaluations in our study were performed using BLUP and a deep pedigree of 29,397 animals that ranged from one to fifteen generations in length (depending on the individual). Additionally, the highest the number of offspring per dam considered in the pedigree, the highest the values of accuracy may be as reported by Iraqui et al. [65]. Our pedigree comprised animals whose offspring ranged from 0 to 846, which may account for the wide range of values and non-normal distribution presented by reliability, accuracy or standard prediction error parameters (Table 4).

Regarding standard errors of prediction (SEP), the widest confidence ranges where reported for milk yield regardless the sex of the animal and whether the genotype for αS1-casein was included or excluded. Despite the inclusion of the genotype did not affect the highest end of the confidence range, the lowest end was higher when genotype was included, a common trend followed by the confidence ranges of fat, protein and dry matter components. The confidence ranges for these components varied very slightly, with the exception of dry matter when genotype was included, which doubled the value of the confidence range when genotype was excluded for both bucks and does. Hence, the risk when making decisions on animal selection is lower for milk yield and dry matter than for the rest of the components when genotype is included. Still, when genotype was considered in the model all traits reported a high enough level of confidence that overcame the values for confidence ranges when genotype was excluded.

High maximum breeding values for milk yield ranged from 137.41 to 156.94, and from 249.04 to 265.12 when αS1 casein was included and excluded, respectively. These high values indicate some breeding does and bucks present a very good genetic potential to improve milk yield, which could be considered when selecting the matings to perform (natural mating or artificial insemination).

When the coefficient of determination (R^2^) for the linear functions described by breeding values for milk yield, protein, fat and dry matter content, were compared between the models including and excluding the genotype of the caseins, they did not exceed the range of 0.0002–0.0052, respectively reported for the production of milk and dry matter when the genotype was included (Figure 1). This finding suggested no significant differences in the breeding values obtained whether the genotype for αS1 casein was included or not. However, these values are not fully consistent with those shown in Table 4 for Pearson correlation coefficients, as despite they hold a highly significant correlation (*p* < 0.001) for breeding values for milk yield and dry matter, it is not high (around 0.061–0.151, respectively), what accounts for the fact that more than linear, this correlation may present a nonlinear nature. Correlations between milk yield and fat or protein contents are conditioned by the fact of including does belonging from different lactational states, hence the low positive or even negative linear correlations reported for such traits and the deviation of the values already reported by literature. Piccardi et al. [66] reported the inclusion of animal random effects, such as in our model, may improve the predictive potential of nonlinear models, but still may not have a significant effect on production indicators such as milk yield, as it was also suggested by the values for heritability obtained in our study resembling those in literature. 

The same finding was reported for the correlations between the breeding values obtained for the fat and protein content (0.049–0.056), respectively. These results, together with those of the determination coefficient (R^2^), suggest the same linear relationship between the breeding values of the traits measured with independence of the inclusion of αS1 genotype in the model.

The PVBs for milk yield are strongly influenced by non-genetic effects, according to Pallete [67], since the environmental factor influences on average up to 70% of milk yield in dairy breeds. On the contrary, dry matter describes the same situation as in the production of milk increasing the reliability and accuracy in the range of 0.11–0.14 (RTi, R_AP_ respectively) for males and 0.18–0.21 (RTi, R_AP_ respectively) for females when the genotype of αS1 casein is not considered as a fixed effect within the model.

Similarly, Carillier-Jacquin et al. [68] suggested αS1-CN genotype may account for a significant effect on milk yield, and composition (fat and protein content) in French dairy goats. These authors reported the fact that considering αS1-casein genotype in genetic and genomic evaluations of bucks αS1 casein genotypes improves the accuracy of the models used in increases that range from 6 to 27%. Similarly, and although it did not reach the same increase in accuracy levels previously reported for males, in genomic evaluations carried out on female records, as in the present study, the accuracy was slightly higher (1% to 14%) than that reported by genomic models which did not involve the genotype for αS1 casein, as also suggested by the slightly higher heritability, precision, and reliability values of estimated breeding values (Table 1 and Table 3).

Genomic selection is a useful tool to complement quantitative genetic evaluation and is currently being implemented and used more and more frequently. Its application needs the cooperation between universities, biotechnology centers, researchers, producers, companies, and breeder associations to work on the development of technologies, and incorporate them into our current evaluation system in order to predict the behavior of future progeny for certain characteristics, especially those of prominent profitability.

## 5. Conclusions

Our findings match and complement those results emerging from studies of genetic polymorphisms of milk proteins in Murciano-Granadina goats given the practical applications derived with the purpose of improving the efficiencies of dairy goat production. Given the different polymorphic forms of milk proteins that are controlled by autosomal genes inherited following Mendelian inheritance patterns, selecting and breeding for a desirable variant within a specific protein is possible by including the effect of genotype in breeding models. The present estimates for genetic parameters enable using milk protein genes as markers for increasing milk yield and improving its content to agree with market demands more efficiently. The effects of certain genetic variants on milk yield and its components, as denoted by low values for BLUES, render mere statistical associations what may be the base for the lack of information about such relationships in literature. Genetic and phenotypic relationships between milk yield and genetic variants of αS1-CN are inconsistently reported in literature due to their reduced reliability stemming from studies accounting for a small population size, the use of different breeds, very low frequencies of some variants, methods of determining yields, and most importantly, the appropriate rigour of statistical procedures used to correct for the major factors and variables, which in turn contributes to higher milk yield ratios. Our study supports the generally addressed tendency for the αS1-CN B allele to be associated with higher milk yield, and for the C variant of αS1-CN, which is associated with higher fat and protein in the milk. The knowledge of the different allelic forms present for the αS1-CN at the molecular level can be of great help to improve the calculation of the genetic value of the breeding animals and above all to serve as a guide to conduct the breeding and genetic selection of animals producing milk with proteins with favourable characteristics for its use in milk and its derivatives (fresh milk, curd, fermentation etc.). Given the implication of the ratio between the fat and protein content in the cheese yield and quality of the curd formation, the inclusion of the genotype as a fixed effect in those models that are used to assess production of goat’s milk and its components can increase the profitability of milk as suggested by the increases in the reliability and accuracy of breeding values for fat and protein, despite the slight decrease in the accuracy of breeding values for other traits such as milk yield and dry matter. However, the negative genetic and phenotypic correlations between milk yield and the content of fat, protein, and dry matter show unfavourable associations, suggesting that the higher the amount of milk produced, the lower its quality is. In line with these results, the positive correlations between fat and protein production, both of which are of great importance for the quality of various dairy products, especially those derived from the cheese industry, suggest the need for the selection of dairy goats based on selection indices that are adapted to these traits considering the genotype of αS1 casein, as they could lead to an increase in the economic value of dairy breeds. The obtained estimates of heritability confirm the existence of a considerable level of genetic diversity that allows the possibility to continue improving the production of milk, fat, protein, and dry matter in breeding and selection programs of Murcia-Granada breed goats.

## Figures and Tables

**Figure 1 animals-09-00679-f001:**
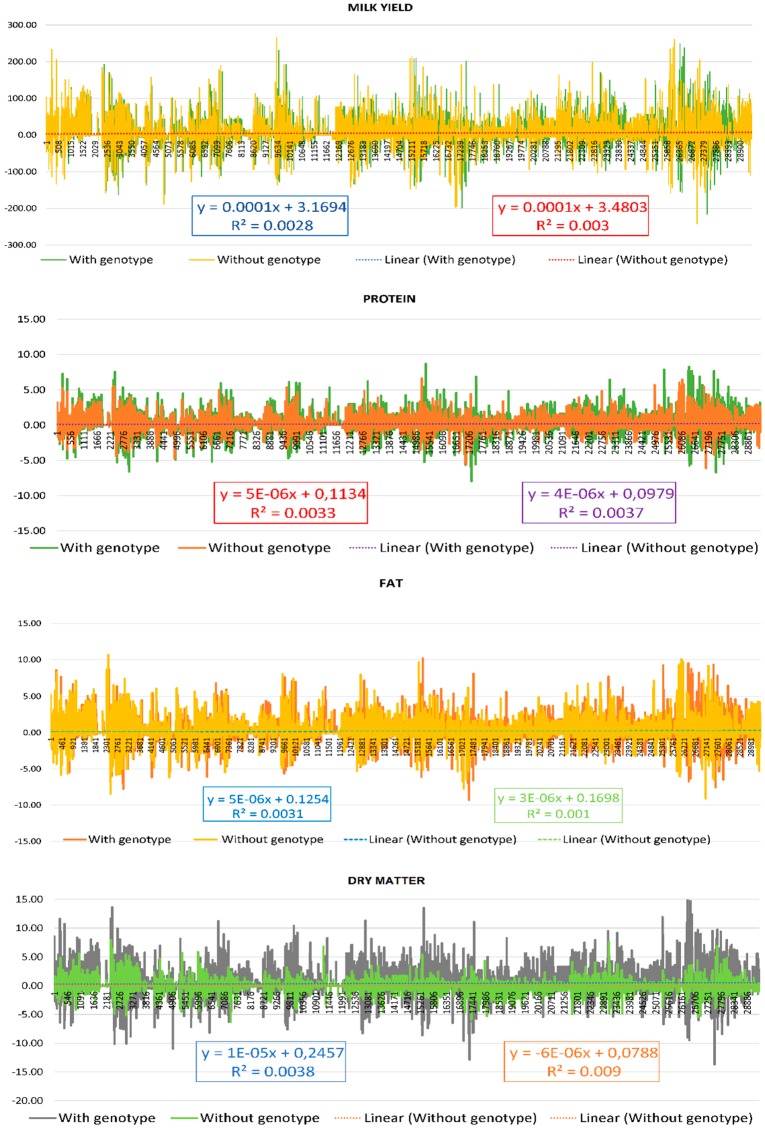
Comparison of predicted breeding values for milk yield or performance, protein, fat and dry matter content in kg, determination coefficients (R^2^) and linear regression equations for their trends obtained using both a model involving the αS1- casein genotype and one excluding it for all the animals included in the pedigree (*n* = 29,397) of Murciano Granadina goat breed.

**Table 1 animals-09-00679-t001:** Kruskal-Wallis H test and partial eta squared (ηp^2^) coefficient results (for milk yield (kg), protein (kg), fat (kg), and dry matter (kg) for Murciano-Granadina goats as described in Pizarro et al. [26].

Factor	Variable (in kg)	χ^2^	P-Value	Dfn,Dfd	F	Partial Eta Squared
Farm	Milk yield	537.74	0.00	58,2031	9.27	0.206
	Protein	457.88	0.00	58,2031	7.89	0.181
	Fat	449.63	0.00	58,2031	7.75	0.178
	Dry matter	455.05	0.00	58,2031	7.85	0.180
Parturition year	Milk yield	115.11	0.00	11,2078	10.46	0.052
	Protein	139.80	0.00	11,2078	12.71	0.062
	Fat	89.13	0.00	11,2078	8.10	0.040
	Dry matter	111.83	0.00	11,2078	10.17	0.050
Parturition month	Milk yield	33.54	0.00	11,2078	3.05	0.016
	Protein	26.66	0.01	11,2078	2.42	0.012
	Fat	40.01	0.00	11,2078	3.64	0.019
	Dry matter	27.41	0.01	11,2078	2.49	0.013
Birth season	Milk yield	20.10	0.00	3,2086	6.70	0.009
	Protein	10.38	0.02	3,2086	3.46	0.005
	Fat	13.21	0.01	3,2086	4.40	0.006
	Dry matter	9.10	0.03	3,2086	3.03	0.004
Birth type	Milk yield	122.25	0.00	4,2085	30.56	0.054
	Protein	116.29	0.00	4,2085	29.07	0.052
	Fat	104.78	0.00	4,2085	26.20	0.047
	Dry matter	110.70	0.00	4,2085	27.68	0.050
Genotype	Milk yield	16.74	0.02	7,2082	2.39	0.008
	Protein	17.12	0.02	7,2082	2.45	0.008
	Fat	19.57	0.02	7,2082	2.80	0.009
	Dry matter	17.44	0.01	7,2082	2.49	0.008
Age	Milk yield	330.97	0.00	92,1997	3.60	0.140
	Protein	326.97	0.00	92,1997	3.55	0.138
	Fat	323.54	0.00	92,1997	3.52	0.137
	Dry matter	325.76	0.00	92,1997	3.54	0.138

dfn: degrees of freedom numerator; dfd: degrees of freedom denominator.

**Table 2 animals-09-00679-t002:** Estimated components of variance, heritability (h^2^) and standard error (SE) for milk yield (kg), protein, fat, and dry matter obtained from multivariate analyses through REML methods in goat milk including and excluding αS1-Casein genotype as a fixed effect.

Model/Genotype as a Fixed Effect	Trait (in kg)	$\sigma_{a}^{2}$	$\sigma_{p}^{2}$	$\sigma_{pe}^{2}$	$\sigma_{e}^{2}$	h^2^ ± SE
Including genotype	Milk yield	11,511.83	28,779.57	1689.51	15,578.23	0.40 ± 0.06
	Fat	16.87	58.83	16.22	25.74	0.29 ± 0.05
	Protein	9.45	17.96	0.36	8.14	0.53 ± 0.02
	Dry matter	57.45	351.07	82.03	211.58	0.16 ± 0.04
Excluding genotype	Milk yield	9525.84	23,814.6	68.74	14,220.02	0.40 ± 0.07
	Fat	14.55	46.57	2.25	29.77	0.31 ± 0.05
	Protein	7.25	24.56	2.99	14.32	0.30 ± 0.03
	Dry matter	49.60	327.25	25.80	251.85	0.15 ± 0.05

**Table 3 animals-09-00679-t003:** Estimated heritabilities (h^2^) (diagonal), phenotypic (r_P_) (above diagonal) and genetic (r_G_) (below diagonal) correlations for milk yield, and protein, fat, and dry matter contents (in kg) obtained in bivariate analyses through REML methods in goat milk including and excluding αS1-Casein genotype as a fixed effect.

Model/Genotype as a Fixed Effect	Trait (kg)	Milk Yield	Fat	Protein	Dry Matter
Including genotype	Milk yield	0.40 ^a^	0.01 ^b^	−0.02 ^b^	0.01 ^b^
	Fat	0.01 ^c^	0.29 ^a^	0.97 ^b^	−0.09 ^b^
	Protein	−0.02 ^c^	0.97 ^c^	0.53 ^a^	0.14 ^b^
	Dry matter	0.01 ^c^	−0.09 ^c^	0.14 ^c^	0.16 ^a^
Excluding genotype	Milk yield	0.40 ^a^	−0.01 ^b^	−0.01 ^b^	0.00 ^b^
	Fat	−0.01 ^c^	0.31 ^a^	0.93 ^b^	0.23 ^b^
	Protein	−0.01 ^c^	0.93 ^c^	0.30 ^a^	−0.15 ^b^
	Dry matter	0.01 ^c^	0.23 ^c^	−0.15 ^c^	0.02 ^a^

^a^ h^2^ ± SE; ^b^ r_P_ ± SE; ^c^ r_G_ ± SE.

**Table 4 animals-09-00679-t004:** Descriptive statistics of predicted breeding values, standard error of prediction (SEP). accuracy (RTi) and reliability (R_AP_) for milk yield (kg). protein. fat. and dry matter obtained in bivariate analyses through REML methods in goat milk including and excluding αS1-Casein genotype as a fixed effect sorted by sex.

Sex	Model	Trait (kg)	Parameter	Minimum	Maximum	Median	Skewness	Kurtosis
Buck (N = 2404)	Including genotype	Milk yield	PBV	−161.54	137.41	0.00	6.76	1.21
			SEP	61.71	119.96	106.86	16.36	−3.17
			RTi	0.00	0.82	0.09	0.35	1.01
			R_AP_	0.00	0.67	0.01	11.09	2.83
		Fat	PBV	−5.37	6.77	0.00	6.71	1.40
			SEP	2.21	4.59	4.09	14.76	−3.09
			RTi	0.00	0.84	0.10	0.24	0.98
			R_AP_	0.00	0.71	0.01	9.71	2.70
		Protein	PBV	−4.88	5.96	0.00	6.39	1.34
			SEP	1.65	3.44	3.06	14.73	−3.08
			RTi	0.00	0.84	0.10	0.24	0.98
			R_AP_	0.00	0.71	0.01	9.77	2.71
		Dry matter	PBV	−10.97	11.22	0.00	5.51	1.14
			SEP	5.61	8.47	7.57	20.98	-2.85
			RTi	0.00	0.67	0.06	0.67	1.08
			R_AP_	0.00	0.45	0.00	16.36	3.23
	Excluding genotype	Milk yield	PBV	−242.51	156.94	0.00	18.49	1.24
			SEP	46.46	109.12	107.29	27.44	−4.92
			RTi	0.00	0.88	0.00	3.55	2.03
			R_AP_	0.00	0.77	0.00	22.49	90.77
		Fat	PBV	−9.09	6.07	0.00	19.55	1.72
			SEP	2.19	4.26	4.11	29.38	12.32
			RTi	0.00	0.82	0.00	3.87	2.09
			R_AP_	0.00	0.67	0.00	23.05	77.45
		Protein	PBV	−6.11	3.78	0.00	17.98	1.77
			SEP	1.69	3.01	3.08	29.01	28.35
			RTi	0.00	0.78	0.00	3.88	2.09
			R_AP_	0.00	0.61	0.00	22.96	76.56
		Dry matter	PBV	−4.60	4.35	0.00	11.71	1.26
			SEP	2.01	3.59	7.58	18.43	3.22
			RTi	0.00	0.78	0.00	2.51	1.87
			R_AP_	0.00	0.61	0.00	17.95	129.83
Doe (N = 26,993)	Including genotype	Milk yield	PBV	−215.76	249.04	0.00	16.26	1.19
			SEP	55.03	119.96	97.60	30.51	−5.14
			RTi	0.00	0.86	0.00	3.52	1.98
			R_AP_	0.00	0.74	0.00	23.74	91.65
		Fat	PBV	−9.30	10.19	0.00	14.94	1.37
			SEP	1.55	4.59	3.81	28.94	39.22
			RTi	0.00	0.93	0.00	3.27	1.95
			R_AP_	0.00	0.86	0.00	23.20	103.84
		Protein	PBV	−7.94	8.65	0.00	12.94	1.57
			SEP	1.17	3.44	2.69	25.88	73.71
			RTi	0.00	0.92	0.00	3.07	1.90
			R_AP_	0.00	0.85	0.00	21.75	106.92
		Dry matter	PBV	−13.68	14.89	0.00	20.59	1.40
			SEP	6.01	8.47	3.21	24.97	49.89
			RTi	0.00	0.61	0.00	3.23	1.98
			R_AP_	0.00	0.37	0.00	21.67	105.64
	Excluding genotype	Milk yield	PBV	−196.97	265.12	0.00	18.49	1.24
			SEP	41.55	109.12	107.29	27.44	−4.92
			RTi	0.00	0.90	0.00	3.55	2.03
			R_AP_	0.00	0.81	0.00	22.49	90.77
		Fat	PBV	−8.44	10.66	0.00	19.55	1.72
			SEP	2.04	4.26	4.11	29.38	12.32
			RTi	0.00	0.85	0.00	3.87	2.09
			R_AP_	0.00	0.72	0.00	23.05	77.45
		Protein	PBV	−5.53	6.65	0.00	17.98	1.77
			SEP	1.67	3.01	3.08	29.01	28.35
			RTi	0.00	0.79	0.00	3.88	2.09
			R_AP_	0.00	0.62	0.00	22.96	76.56
		Dry matter	PBV	−6.27	7.92	0.00	11.71	1.26
			SEP	1.98	3.59	7.58	18.43	3.22
			RTi	0.00	0.79	0.00	2.51	1.87
			R_AP_	0.00	0.62	0.00	17.95	129.83

**Table 5 animals-09-00679-t005:** Pearson product-moment correlation coefficients between the predicted breeding values (PBV) for milk yield and fat, protein and dry matter (in kg) obtained using both a model involving the αS1- casein genotype and one excluding it for all the animals included in the pedigree (*n* = 29,397) of Murciano Granadina goat breed.

Milk Yield	Fat	Protein	Dry Matter
0.061 **	0.049 **	0.056 **	0.151 **

** *p* < 0.001.

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
