# Peer review of "Does the Acknowledgement of αS1-Casein Genotype Affect the Estimation of Genetic Parameters and Prediction of Breeding Values for Milk Yield and Composition Quality-Related Traits in Murciano-Granadina?"

_animals, 2019, doi:10.3390/ani9090679_

Round 1

Reviewer 1 Report

General comments

The study compared PBV with and without genotypes. The results of all traits show very large differences (low R2 and correlations) between including genotypes and excluding genotypes. Assuming that PBV including genotypes are accurate, those genotypes control almost everything in genetics for all traits. Authors need to focus on explaining why the results are so different by including genotype effects.

I suggest adding one table including effects of genotypes for all traits.

Introduction

Lines 81-87: How much is the phenotypic percentage of αS1-Casein in the total amount of proteins?

Line 93: What is “milk performance”, 210-d milk yield?

Materials and Methods

Lines 284-285: Usually, “BLUES” is used for fixed effects, and “BLUPS” is used for random effects. So, all estimates of fixed effects are BLUE, and all predictions of random effects (animal and PE in this case) are BLUP.

Results

Table 2: If the difference between the 2 models is only including and excluding genotypes, variances should not be very different. The PE variances are too different. Usually, a model including more fixed effects should have the smaller total variance.

Table 3: I still don’t get what kind of trait the milk performance is. Authors sometimes use “performance”, “production”, or “yield”, but I guess that they mean the same (210-d milk yield) as well as fat yield and protein yield. Genetic and phenotypic correlations of milk with other traits should not be very low because they are part of milk yield (milk=fat+protein+….).

Table 4: Explain why so important to show those statistics (minimum, maximum, median, …..). SEP, reliability, and accuracy are basically the same information, using different scale.

What does ”zero reliability” for PBV mean? If an animal has phenotype(s), the reliability should be > zero. If an animal has parent(s) with breeding values, the reliability should be > zero. If an animal has progeny with phenotypes, the reliability should be > zero.

Table 5: The table needs only one row or one column to show those correlations.

Table 5 and Figure 1: Even for traits other than protein, R2 and correlations between PBV with and without genotypes are too low (close to zero), indicating that something must be wrong. Assuming that PBV with genotypes are more accurate, PBV without genotypes are useless.

Author Response

All the team responsible for this paper acknowledge the comments from the reviewers and editor, as they help to improve the quality of our manuscript. In the following paragraphs, we will describe and address how the referees’ recommendations were followed. A point-by-point response to comments is provided as well as a file where changes are highlighted.

Reviewer 1

The study compared PBV with and without genotypes. The results of all traits show very large differences (low R2 and correlations) between including genotypes and excluding genotypes. Assuming that PBV including genotypes are accurate, those genotypes control almost everything in genetics for all traits. Authors need to focus on explaining why the results are so different by including genotype effects.

Response: Partial eta squared values report a very low, still significant association between αS1 casein and the traits studied in the manuscript (in the range of 0.8 to 0.9%). This may account for the low R2 and low correlations associated to the inclusion/exclusion of the genotype as a fixed effect. Following reviewer suggestions, a lot of information is provided through the text and has been further explained as a way to improve clarity and to further explain why results may be different, though not generally.

I suggest adding one table including effects of genotypes for all traits.

Response: Lines 408 to 411. Effects for genotypes for all traits have been included in Supplementary Table S4 among the values provided for the BLUES. These results, describing the existing deviation for the mean respecting the different levels of each certain fixed effect (in this case, the genotype for αS1 casein) support our findings as it can be learnt from discussion in lines 566-567.

Introduction

Lines 81-87: How much is the phenotypic percentage of αS1-Casein in the total amount of proteins?

Response: Line 88. Percentage of αS1-Casein in the total amount of proteins for Murciano Granadina breed was included as well as a reference from which the data was extracted. All references were recoded afterwards to follow the order according to journal guidelines.

Line 93: What is “milk performance”, 210-d milk yield?

 Response: Yes, we changed it through the manuscript to improve clarity.

Materials and Methods

Lines 284-285: Usually, “BLUES” is used for fixed effects, and “BLUPS” is used for random effects. So, all estimates of fixed effects are BLUE, and all predictions of random effects (animal and PE in this case) are BLUP.

Response: True, exactly as it is described in this paper. However, we clarified from lines 294-295.

Results

Table 2: If the difference between the 2 models is only including and excluding genotypes, variances should not be very different. The PE variances are too different. Usually, a model including more fixed effects should have the smaller total variance.

Response. Variances do not differ very much, even more if we consider the error reported for heritabilities. Anyway, smaller total variances should be expected in those models including more fixed effects but when there has not been a specific selection process for the sample as the one described in Pizarro et al. 2019 on page 1023 << For these reasons, we chose 236 females presenting the lowest ICO values in the rank, 238 females with values around percentile 50, and the 236 females presenting the highest ICO values in the rank, so as to perform an adjusted representative sampling of the genotype distribution in the population>> as it is reported in our paper in line 124.

Table 3: I still don’t get what kind of trait the milk performance is. Authors sometimes use “performance”, “production”, or “yield”, but I guess that they mean the same (210-d milk yield) as well as fat yield and protein yield.

Response: We made the nomenclature for milk yield homogeneous through the whole manuscript to improve readability and avoid misconceptions.

Genetic and phenotypic correlations of milk with other traits should not be very low because they are part of milk yield (milk=fat+protein+….).

 Response: Literature is full of examples reporting milk yield holds a variably negative correlation with milk content, specially, fat and protein, across species (cows, sheep and goats). This does not happen with lactose as reviewer may already know for which the event described in the reviewer’s comment occurs, the more lactose the more milk (osmotic attraction of water). However, lactose was not included in our study. You my find example of references accounting for this event through discussion.

Table 4: Explain why so important to show those statistics (minimum, maximum, median, …..). SEP, reliability, and accuracy are basically the same information, using different scale.

Response: Descriptive statistics are important, especially those reported in table 4 in non-normally distributed variable cases, as they provide a better perspective of the distribution of values for such variable across a certain population, while other like the mean would not be appropriate. Then, there is another misconception. SEP, reliability and accuracy do not provide basically the same information. They differ from the very definition. On the one hand, reliability could be described as the likeliness of someone repeating the experiment and getting the same result (repeatability), whereas accuracy is how close your value is to the real value, hence values should be interpreted differently. On the other hand, The Standard prediction error measures how large prediction errors (residuals) are for your data set measured in the same unit as your variable. Then statistically, if these parameters were the same using a different scale skewness and kurtosis values showing the dispersion of values could presumably be expected not to differ, but they do, what statistically supports the differences in definitions. We do not think it is important to add this information in the text as these concepts are widely applied and known for the research discipline with which this paper deals. Manuals for the software used have been referenced hence readers could consult such materials in case of need for further explanations.

What does ”zero reliability” for PBV mean? If an animal has phenotype(s), the reliability should be > zero. If an animal has parent(s) with breeding values, the reliability should be > zero. If an animal has progeny with phenotypes, the reliability should be > zero.

Response: close to zero reliabilities may account for an animal whose parents (one or both) are founders in the population (That is from which no ancestor is known), but for whom a valid phenotypic value may have been recorded. This information was clarified as a footnote in the Table.

Table 5: The table needs only one row or one column to show those correlations.

Response: Table was formatted following reviewer’s suggestions.

Table 5 and Figure 1: Even for traits other than protein, R2 and correlations between PBV with and without genotypes are too low (close to zero), indicating that something must be wrong. Assuming that PBV with genotypes are more accurate, PBV without genotypes are useless.

 Response: Our results suggest, as it has been stated through discussion, an increase in reliability when we include the genotype among the fixed factors comprising our model (as it had also been suggested by genetic parameters estimates) what could account for the slight distortion occurring in R2 values and for the relatively low though still highly statistically significant correlations for predicted breeding values. However, we are concerned about reviewer thinking less accurate may mean something is useless. For example, in low budget cases when animal selection practices want to be carried out but there is no sufficient money as to implement more complex and expensive practices as to have a rather inaccurate value is better than not to have any. Hence, this information is important, especially for some parts of the world.

Reviewer 2 Report

Manuscript Animals-579172, entitled “Does knowing αS1-casein genotype affect the estimation of genetic parameters and prediction of breeding values for milk yield and composition quality related traits in Murciano-Granadina?”

Recommendation: The above paper is not suitable for publication in its present form.

General Comments:

The article provides useful information about the utilization of αS1-casein genotype in the prediction of breeding values for milk yield and composition quality related traits in Murciano-Granadina goats. Although the experiment is appropriately designed and implemented, there are a lot of grammar, stylistic and syntax errors. In some cases, these errors negatively influence the understanding of the text.

Another part that needs improvement is the general conclusion of the authors. According to authors (L33-34), heritability is improved only for milk protein production after aS1-CN genotype is included in the breeding model. No significant differences were observed for milk yield, fat and dry matter. Please rephrase accordingly.

Some minor points should also be corrected.

Specific comments

Title: “Does acknowledgement of aS1-casein genotype…”

L20: “…for economically important traits. As a result, dairy livestock have…”

L21-22: “…in goats has been mostly based on quantitative…”

L23: “…reported by increased heritability values of these parameters after the inclusion…”

L24-25: “…evaluation models, the selection of animals carrying this gene could result in a more efficient genetic selection. High levels of genetic…”

L26: “are” instead of “that would be”

L27-28: “…proteins in milk, a fact that could be associated with higher yield and improved curd firmness properties.”

L29-30: “…were collected during the years 2005-2016…”

L32: “…assess its repercussion on the efficiency…”

L35: “that the consideration of” instead of “considering”

L39: “…reliability -Rap- (0.45-0.71) was similar to that of females (0.37-0.86) when…”

L42-43: Please check general comment

L49: “…is the most popular breed in Spain and it is distributed across…”

L56: “…of goat derived products, such as milk, cheese and meat, has been increased by…”

L57-58: “…over the last 20 years [5] as a result of the consumers’ preferences. Not only goats…”

L60: Please delete “for example,”

L61-62: “Genetic selection has contributed in the development and advancement of dairy sector, focusing on the improvement of economically important aspects associated with increased productivity…”

L67: “result in” instead of “help”

L73: “in many” instead of “for very distant”

L75: “but also from” instead of “to”

L113-115: Please rephrase

L117: “a fact that” instead of “what”

L119: What had been accomplished?

L120: “…during the years 2005-2016…”

L122-124: “…were performed aiming at the prevention of incorrect as an effect of the epistasis between traits) for the selection of the animals that could be genotyped. Then, we applied mixed model procedures…”

L137 and throughout the text: Please delete “,” before “et al.”

L139: “…of milk yield, protein, fat…”

L140: “approach”

L141: “Given”?

L146: Please delete “going”

L159: “Brito et al. [28]”

L160-161, 173-174, 177-178, 190-191, 194-195, 205-206, 248-249, 337-338: “Pizarro et al. [27]”

L165-166: “…official Milk Quality Laboratory (Córdoba, Spain) for composition determination…”

L178: “…from 117225 blood samples…”

L201: Please delete “present”

L206: “…described by these authors…”

L208: “a previous study” instead of “this paper”

L210-213: Please rephrase

L215: “to examine whereas” instead of “given”

L222: What do you mean by “staistically significan”?

L225-226: Please rephrase

L244: “…was performed…”

L254: “…records of 29397 animals…”

L257: “examination” instead of “study”

L261: “notation, the following multitrait animal model with repeated measures was used:”

L284-295: Repetition. Please shorten this paragraph.

L306: Please delete “from both models”

L314: “special”

L319-321: Please rephrase

L327-328: Please rephrase

L332: “This relies on…”

L337: “found” instead of “consulted”

L357 and throughout the text: Please use “bucks” instead of “billies” and “does” instead of “goats”

L362: Please delete “for goats”

Table 5: What do you mean by “**”?

L388-389: “…of αS1-CN genotype, considering that genotypic variants…”

L393: “…or yoghurt production, the profitability of dairy goat breeding programs can be optimized [37].”

L414: “On the other hand,” instead of “Whereas”

L421: “Analla et al. [42].”

L424-425: “(0.17 to 0.41) [43-45]; fat (0.26 to 0.62) [46]; protein (0.14 to 0.67) [40,46] and dry matter (0.16 to 0.36) [47].”

L428-429: “…breed goats, even although they is a highly selected breed [28,45,47,48].”

L455-456: Please rephrase

L468: “…between the fat and protein components…”

L482: “…calculating it as fresh…”

L487: “contribute in an improved” instead of “translate in a better”

L488-489: “Pizarro et al. [27] suggested that such higher fat/protein ration could be possibly reported in goats presenting…”

L500: “…from females were available, that from males…”

L512: “…values are high by including or excluding the…”

L513-515: Please rephrase

L516: “obtention”?

L527: “states” instead of “statuses”

L538: “Pallete (2001)” Please check reference style.

L538: “…Pallete (2001), since the environmental factor influences on average up to 70% of milk production in dairy…”

L540-541: “…for males and 0.18-0.21 (RTi, Rap respectively) for females…”

L552: “…evaluation and is…”

L553: “…more frequently. Its application needs the cooperation between…”

L567-568: Please rephrase

L573: “contribute” instead of “translate”

L582: “…models that are used to assess production…”

L583: Please delete “result very beneficial to”

L593: “…spite of all, the obtained estimates of heritability confirm…”

Author Response

All the team responsible for this paper acknowledge the comments from the reviewers and editor, as they help to improve the quality of our manuscript. In the following paragraphs, we will describe and address how the referees’ recommendations were followed. A point-by-point response to comments is provided as well as a file where changes are highlighted.

Reviewer 2

Comments and Suggestions for Authors

Manuscript Animals-579172, entitled “Does knowing αS1-casein genotype affect the estimation of genetic parameters and prediction of breeding values for milk yield and composition quality related traits in Murciano-Granadina?”

Recommendation: The above paper is not suitable for publication in its present form.

General Comments:

The article provides useful information about the utilization of αS1-casein genotype in the prediction of breeding values for milk yield and composition quality related traits in Murciano-Granadina goats. Although the experiment is appropriately designed and implemented, there are a lot of grammar, stylistic and syntax errors. In some cases, these errors negatively influence the understanding of the text.

Response: The whole text was rechecked seeking to correct for grammar, stylistic and syntax errors according to reviewer’s suggestion.

Another part that needs improvement is the general conclusion of the authors. According to authors (L33-34), heritability is improved only for milk protein production after aS1-CN genotype is included in the breeding model. No significant differences were observed for milk yield, fat and dry matter. Please rephrase accordingly.

Response: Rephrased according to reviewer’s suggestion.

Some minor points should also be corrected.

Specific comments

Title: “Does acknowledgement of aS1-casein genotype…”

Response: Changed according to reviewer’s suggestion.

L20: “…for economically important traits. As a result, dairy livestock have…”

Response: Changed according to reviewer’s suggestion.

L21-22: “…in goats has been mostly based on quantitative…”

Response: Changed according to reviewer’s suggestion.

L23: “…reported by increased heritability values of these parameters after the inclusion…”

Response: Changed according to reviewer’s suggestion.

L24-25: “…evaluation models, the selection of animals carrying this gene could result in a more efficient genetic selection. High levels of genetic…”

Response: Changed according to reviewer’s suggestion.

L26: “are” instead of “that would be”

Response: Changed according to reviewer’s suggestion.

L27-28: “…proteins in milk, a fact that could be associated with higher yield and improved curd firmness properties.”

Response: Changed according to reviewer’s suggestion.

L29-30: “…were collected during the years 2005-2016…”

Response: Changed according to reviewer’s suggestion.

L32: “…assess its repercussion on the efficiency…”

Response: Changed according to reviewer’s suggestion.

L35: “that the consideration of” instead of “considering”

Response: Changed according to reviewer’s suggestion.

L39: “…reliability -Rap- (0.45-0.71) was similar to that of females (0.37-0.86) when…”

Response: Changed according to reviewer’s suggestion.

L42-43: Please check general comment

Response: We rewrote the information as suggested by the reviewer in his/her general comment.

L49: “…is the most popular breed in Spain and it is distributed across…”

Response: Changed according to reviewer’s suggestion.

L56: “…of goat derived products, such as milk, cheese and meat, has been increased by…”

Response: Changed according to reviewer’s suggestion.

L57-58: “…over the last 20 years [5] as a result of the consumers’ preferences. Not only goats…”

Response: Changed according to reviewer’s suggestion.

L60: Please delete “for example,”

Response: Deleted according to reviewer’s suggestion.

L61-62: “Genetic selection has contributed in the development and advancement of dairy sector, focusing on the improvement of economically important aspects associated with increased productivity…”

Response: Changed according to reviewer’s suggestion.

L67: “result in” instead of “help”

Response: Changed according to reviewer’s suggestion.

L73: “in many” instead of “for very distant”

Response: Changed according to reviewer’s suggestion.

L75: “but also from” instead of “to”

Response: Changed according to reviewer’s suggestion.

L113-115: Please rephrase

Response: Sentence was rephrased according to reviewer’s suggestions.

L117: “a fact that” instead of “what”

Response: Changed according to reviewer’s suggestion.

L119: What had been accomplished?

Response: Information has been clarified.

L120: “…during the years 2005-2016…”

Response: Changed according to reviewer’s suggestion.

L122-124: “…were performed aiming at the prevention of incorrect as an effect of the epistasis between traits) for the selection of the animals that could be genotyped. Then, we applied mixed model procedures…”

Response: Changed according to reviewer’s suggestion.

L137 and throughout the text: Please delete “,” before “et al.”

Response: Deleted according to reviewer’s suggestion.

L139: “…of milk yield, protein, fat…”

Response: Changed according to reviewer’s suggestion.

L140: “approach”

Response: Changed according to reviewer’s suggestion.

L141: “Given”?

Response: Clarified.

L146: Please delete “going”

Response: Deleted according to reviewer’s suggestion.

L159: “Brito et al. [28]”

Response: Changed according to reviewer’s suggestion.

L160-161, 173-174, 177-178, 190-191, 194-195, 205-206, 248-249, 337-338: “Pizarro et al. [27]”

Response: Changed according to reviewer’s suggestion.

L165-166: “…official Milk Quality Laboratory (Córdoba, Spain) for composition determination…”

Response: Changed according to reviewer’s suggestion.

L178: “…from 117225 blood samples…”

Response: Changed according to reviewer’s suggestion.

L201: Please delete “present”

Response: Changed according to reviewer’s suggestion.

L206: “…described by these authors…”

Response: Changed according to reviewer’s suggestion.

L208: “a previous study” instead of “this paper”

Response: Changed according to reviewer’s suggestion.

L210-213: Please rephrase

Response: Rephrased according to reviewer’s suggestion.

L215: “to examine whereas” instead of “given”

Response: Changed according to reviewer’s suggestion.

L222: What do you mean by “staistically significan”?

Response: Statistically significant means results were statistically significant as denoted by p-values.

L225-226: Please rephrase

Response: Rephrased according to reviewer’s suggestion.

L244: “…was performed…”

Response: Changed according to reviewer’s suggestion.

L254: “…records of 29397 animals…”

Response: Changed according to reviewer’s suggestion.

L257: “examination” instead of “study”

Response: Changed according to reviewer’s suggestion.

L261: “notation, the following multitrait animal model with repeated measures was used:”

Response: Changed according to reviewer’s suggestion.

L284-295: Repetition. Please shorten this paragraph.

Response: Changed according to reviewer’s suggestion. Paragraph was shortened and the information provided was clarified.

L306: Please delete “from both models”

Response: Deleted according to reviewer’s suggestion.

L314: “special”

Response: Changed according to reviewer’s suggestion.

L319-321: Please rephrase

Response: Rephrased according to reviewer’s suggestion.

L327-328: Please rephrase

Response: Rephrased according to reviewer’s suggestion.

L332: “This relies on…”

Response: Changed according to reviewer’s suggestion.

L337: “found” instead of “consulted”

Response: Changed according to reviewer’s suggestion.

L357 and throughout the text: Please use “bucks” instead of “billies” and “does” instead of “goats”

Response: Changed according to reviewer’s suggestion.

L362: Please delete “for goats”

Response: Deleted according to reviewer’s suggestion.

Table 5: What do you mean by “**”?

Response: Clarified as a footnote according to reviewer’s suggestion.

L388-389: “…of αS1-CN genotype, considering that genotypic variants…”

Response: Changed according to reviewer’s suggestion.

L393: “…or yoghurt production, the profitability of dairy goat breeding programs can be optimized [37].”

Response: Changed according to reviewer’s suggestion.

L414: “On the other hand,” instead of “Whereas”

Response: Changed according to reviewer’s suggestion.

L421: “Analla et al. [42].”

Response: Changed according to reviewer’s suggestion.

L424-425: “(0.17 to 0.41) [43-45]; fat (0.26 to 0.62) [46]; protein (0.14 to 0.67) [40,46] and dry matter (0.16 to 0.36) [47]

Response: Changed according to reviewer’s suggestion.

L428-429: “…breed goats, even although they is a highly selected breed [28,45,47,48].”

Response: Suggestion by reviewer was grammatically incorrect. This is our proposal to accommodate reviewer’s suggestion in a joint sentence.

L455-456: Please rephrase

Response: Rephrased according to reviewer’s suggestion.

L468: “…between the fat and protein components…”

Response: Changed according to reviewer’s suggestion.

L482: “…calculating it as fresh…”

Response: Changed according to reviewer’s suggestion.

L487: “contribute in an improved” instead of “translate in a better”

Response: Changed according to reviewer’s suggestion.

L488-489: “Pizarro et al. [27] suggested that such higher fat/protein ration could be possibly reported in goats presenting…”

Response: Changed according to reviewer’s suggestion.

L500: “…from females were available, that from males…”

Response: Changed according to reviewer’s suggestion.

L512: “…values are high by including or excluding the…”

Response: Changed according to reviewer’s suggestion.

L513-515: Please rephrase

Response: Rephrased according to reviewer’s suggestion.

L516: “obtention”?

Response: Rephrased, as the sentence was not clear.

L527: “states” instead of “statuses”

Response: Changed according to reviewer’s suggestion.

L538: “Pallete (2001)” Please check reference style.

Response: Checked according to reviewer’s suggestion.

L538: “…Pallete (2001), since the environmental factor influences on average up to 70% of milk production in dairy…”

Response: Changed according to reviewer’s suggestion.

L540-541: “…for males and 0.18-0.21 (RTi, Rap respectively) for females…”

Response: Changed according to reviewer’s suggestion.

L552: “…evaluation and is…”

Response: Changed according to reviewer’s suggestion.

L553: “…more frequently. Its application needs the cooperation between…”

Response: Changed according to reviewer’s suggestion.

L567-568: Please rephrase

Response: Rephrased, as the sentence was not clear.

L573: “contribute” instead of “translate”

Response: Changed according to reviewer’s suggestion.

L582: “…models that are used to assess production…”

Response: Changed according to reviewer’s suggestion.

L583: Please delete “result very beneficial to”

Response: Deleted according to reviewer’s suggestion.

L593: “…spite of all, the obtained estimates of heritability confirm…”

Response: Changed according to reviewer’s suggestion.

Round 2

Reviewer 1 Report

Table 3: I have been working on dairy cattle for 40 years, but I have seen very few papers showing zero or negative (genetic) correlations of milk yield with fat yield or protein yield (of course it is possible with fat % or protein %). They are usually 0.5 to 0.9. Maybe goats are different.

 Table 4: What I mentioned was that reliability = accuracy^2 = (1 - SEP^2/V(a))^2. Each interpretation must be different.

Zero reliability (accuracy) can happen only when the animal has no phenotypes and no parents. If the parents have no phenotypes and no progeny, they should not be included in the analysis. Even if founder parents have no phenotypes but have progeny with phenotypes, their reliabilties are not zero.

Author Response

Francisco Javier Navas González

Department of Genetics, Faculty of Veterinary Sciences

University of Córdoba

Rabanales University Campus, 14071 Córdoba (Spain)

+34 651679262

[email protected]

08/09/2019

Dear Editor,

All the team responsible for this paper acknowledge the comments from the reviewer and editor, as they help to improve the quality of our manuscript. In the following paragraphs, we will describe and address how referee’s new recommendations were followed. A point-by-point response to comments is provided as well as a file where changes are highlighted.

Reviewer 1

Comments and Suggestions for Authors

Table 3: I have been working on dairy cattle for 40 years, but I have seen very few papers showing zero or negative (genetic) correlations of milk yield with fat yield or protein yield (of course it is possible with fat % or protein %). They are usually 0.5 to 0.9. Maybe goats are different.

Response: We agree with the reviewer. We feel there may have been a misunderstanding. From lines 202 and 210, we describe how determination of components was performed as percentages and later multiplied by the whole value for milk yield in Kgs. This may account for the existence of negative correlations, as not all animals have the same percentage of protein or fat, what may distort the causative effect described by the reviewer in the first round (more milk=more protein/more fat). We have tried to find examples of milk yield to components correlations (in Kgs) rather than percentages for goats in literature and the results are scarce, as the study of percentages as the one carried out in our study is the preferable option.

We added the following sentence from lines 428 to 430 to clarify ”These negative correlations have been widely reported to be a result of the procedure followed to quantify milk composition using percentages or fractions (g/kg)”

Furthermore, the existence of a negative linkage disequilibrium between loci may account for such negative genetic relationships between milk yield and content traits as described from line 458 to 460 “These negative values have been reported for genomic correlations between loci and have been attributed to negative linkage disequilibrium expected for traits under directional selection [52] as it would happen in our study.”

Table 4: What I mentioned was that reliability = accuracy^2 = (1 - SEP^2/V(a))^2. Each interpretation must be different.

Response: Following reviewer we added further information to clarify the differences in the interpretation of the three parameters.

“Standard error of Prediction (SEP), reliability (RAP) and accuracy (RTi) do not provide basically the same information. They differ from the very definition and equation determination (RAP = RTi2 = (1 - SEP2/Va)2, from which Va is genetic additive variance. On the one hand, reliability could be described as the likeliness of someone repeating the experiment and getting the same result (repeatability), whereas accuracy is how close your value is to the real value, hence values should be interpreted differently. According to the International Beef Recording Scheme [36]. The following guide may assist as a rule of thumb to interpret accuracy (RAP). Less than 50% accuracies mean PBVs are preliminary, thus calculated basing on little information and hence very prone to change substantially as more direct information on the animal becomes available. Accuracies ranging from 50-74% accuracy (medium) suggests PBVs may have been calculated based on the animal direct information and some limited indirect pedigree information. Medium/high accuracies are denoted by 75-90% and may be calculated considering the animal’s direct information together with the performance of a small number of its offspring. Accuracy values over 90%, report estimates of the animal’s true breeding value, as it unlikely that PBVs will change considerably even if additional information from offspring is added. Regarding reliability, the rule of thumb proposed by KWPN (Royal Dutch Sport Horse) [37] is as follows; values less than 30% are generally unreliable, 30-55% poor reliability, 55-65% sufficient reliability, 65-75% more than sufficient reliability, 75-90% good reliability, >90% very reliable and repeatable with values around 60% meaning the information strongly relies of offspring information, what would not be desirable. On the other hand, the standard error of prediction (SEP) measures how large prediction errors (residuals) are for your data set measured in the same unit as your variable, hence provides a direct measure of possible change, that is the risk of the true breeding value of animal (TBV) not to be centered on the PBV. According to Van Vleck [38], possible change is risk in units of the trait and can be ‘positive’ or ‘negative’, what means the chance true BV may exceed PBV by a certain amount (possible gain) is the same as the chance true BV is less than PBV by the same amount (possible loss). In this context, confidence ranges are frequently used to determine probabilities of possible change assuming a normal distribution of TBV around the PBV. One-half of TBV would be expected to be greater than the PBV and one-half would be expected to be less than the PBV. The interval from PBV – (1)SEP to PBV + (1)SEP corresponds to 68% of likelihood that the TBV for an animal is centered on the PBV for the animal. The range can be narrowed or widened corresponding to the probability of TBV in the interval. For example, the interval from PBV – (2)SEP to PBV + (2)SEP would be expected to contain 95% of TBV. Units of SEP other than (1) or (2) would correspond to other confidence ranges. With a 68% confidence range, 32% would be half over and half below the ranges’ ends, while with the 95% range, the percentage falling out of the range would be 5% (again half over and half below each end, respectively). Ranges for many combinations of PBV and SEP may overlap considerably. Then observing which PBV centers the range and comparing ranges we may obtain a more direct measure of risk than that from accuracy (RTi).”

And from lines 552 to 554: This is supported by RTi scores, which suggest that PBVs may have been calculated based on the animal direct information and some limited indirect pedigree ancestral information or belonging to a small number of its offspring.

And from lines 587 to 597: “Regarding standard errors of prediction (SEP), the widest confidence ranges where reported for milk yield regardless the sex of the animal and whether the genotype for αS1-casein was included or excluded. Despite the inclusion of the genotype did not affect the highest end of the confidence range, the lowest end was higher when genotype was included, a common trend followed by the confidence ranges of fat, protein and dry matter components. The confidence ranges for these components varied very slightly, with the exception of dry matter when genotype was included, which doubled the value of the confidence range when genotype was excluded for both bucks and does. Hence the risk to take when making decisions on animal selection is lower for milk yield and dry matter than for the rest of the components when genotype is included. Still, when genotype was considered in the model all traits reported a high enough level of confidence that overcame the values for confidence ranges when genotype was excluded.”

Zero reliability (accuracy) can happen only when the animal has no phenotypes and no parents. If the parents have no phenotypes and no progeny, they should not be included in the analysis. Even if founder parents have no phenotypes but have progeny with phenotypes, their reliabilties are not zero.

Response: We understand the reviewer’s point and agree with him, however, that was not what we wanted to express and we may have not explained in a good manner the previous round. According to Clark et al. 2012, how distant the animals are in deep informative pedigrees may play a role in the appearance of close to zero or even zero accuracies. We included the following paragraph to clarify.

“Some authors have reported the possibility of accuracies to reach values close to zero or even zero for distant or unrelated animals when genetic analyses are performed on deep pedigrees (that is considering a large number of generations, involving a great number of unknown genetic background animals -founders-) as the one in our study (from 1 to 15 depending on the animal). Clark et al. [62] calculated accuracy as the correlation between estimated and true breeding values for 250 animals with no phenotype ranging from 1 to 8 generations (depending on the individual). These authors did not find significant differences in accuracy between shallow or deep pedigree BLUPS when the animals in the pedigree had a close relationship. However, when a shallow pedigree was used, and animals were distantly related (maximum relationship ranging from 0 to 0.125) or unrelated (no pedigree relationship), all breeding values estimated were zero and accuracies close to zero (Average relatedness in our pedigree ranged from 0 to 1.07%). In contrast, when pedigree was deep, breeding values were predicted with a significant accuracy when there was a distant relationship between animals and accuracy reduced to very close to zero when animals were unrelated, what may support our findings (Table 4). Genetic evaluations in our study were performed using BLUP and a deep pedigree of 29397 animals that ranged from one to fifteen generations in length (depending on the individual). Additionally, the highest the number of offspring per dam considered in the pedigree, the highest the values of accuracy may be as reported by Iraqui et al. [63]. Our pedigree comprised animals whose offspring ranged from 0 to 846, what may account for the wide range of values and non-normal distribution presented by reliability, accuracy or standard prediction error parameters (Table 4).”

Should you require any further information do not hesitate to contact me via email or telephone at any time.

Yours faithfully,

Dr Francisco Javier Navas González
